

# Decoding Bitcoin: leveraging macro- and micro-factors in time series analysis for price prediction

Hae Sun Jung[1], Jang Hyun Kim[2] and Haein Lee[3]

[1] Department of Applied Artificial Intelligence, Sung Kyun Kwan University, Seoul, Republic of South Korea
[2] Department of Interaction Science/Department of Human-Artificial Intelligence Interaction, Sung Kyun Kwan University, Seoul, Republic of South Korea
[3] Department of Applied Artificial Intelligence/Department of Human-Artificial Intelligence Interaction, Sung Kyun Kwan University, Seoul, Republic of South Korea

## ABSTRACT

Predicting Bitcoin prices is crucial because they reflect trends in the overall cryptocurrency market. Owing to the market's short history and high price volatility, previous research has focused on the factors influencing Bitcoin price fluctuations. Although previous studies used sentiment analysis or diversified input features, this study's novelty lies in its utilization of data classified into more than five major categories. Moreover, the use of data spanning more than 2,000 days adds novelty to this study. With this extensive dataset, the authors aimed to predict Bitcoin prices across various timeframes using time series analysis. The authors incorporated a broad spectrum of inputs, including technical indicators, sentiment analysis from social media, news sources, and Google Trends. In addition, this study integrated macroeconomic indicators, on-chain Bitcoin transaction details, and traditional financial asset data. The primary objective was to evaluate extensive machine learning and deep learning frameworks for time series prediction, determine optimal window sizes, and enhance Bitcoin price prediction accuracy by leveraging diverse input features. Consequently, employing the bidirectional long short-term memory (Bi-LSTM) yielded significant results even without excluding the COVID-19 outbreak as a black swan outlier. Specifically, using a window size of 3, Bi-LSTM achieved a root mean squared error of 0.01824, mean absolute error of 0.01213, mean absolute percentage error of 2.97%, and an R-squared value of 0.98791. Additionally, to ascertain the importance of input features, gradient importance was examined to identify which variables specifically influenced prediction results. Ablation test was also conducted to validate the effectiveness and validity of input features. The proposed methodology provides a varied examination of the factors influencing price formation, helping investors make informed decisions regarding Bitcoin-related investments, and enabling policymakers to legislate considering these factors.

Corresponding author
Haein Lee, lhi00034@g.skku.edu

# INTRODUCTION

Although debates on the merits of blockchain technology and the value of cryptocurrencies persist, the cryptocurrency market has grown substantially, attracting both institutional and individual investors (*Mikhaylov, 2020*; *Perchuk, Makarchuk & Yaremenko, 2019*; *Jung, Lee & Kim, 2023a*). Recent figures indicate that the cryptocurrency market's capitalization has surpassed 1.1 trillion dollars (*Finneseth, 2022*). Consequently, cryptocurrencies, including Bitcoin, are increasingly recognized as a distinct asset class, prompting the gradual establishment of cryptocurrency-related regulations (*Jung et al., 2023b*).

Prior research has indicated that cryptocurrencies such as Bitcoin can serve portfolio diversification and risk management (*Bouri et al., 2017*; *Alarab & Prakoonwit, 2023*; *Corbet et al., 2018*). As a prominent player in the cryptocurrency market, Bitcoin has played a crucial role since its inception on 2008 (*Nakamoto, 2008*). Previous research also suggested that other cryptocurrencies often follow Bitcoin's price trends (*Ciaian & Rajcaniova, 2018*). Therefore, predicting Bitcoin price trends can provide insights into market movements for cryptocurrency investors and policymakers. Moreover, understanding the factors that impact the formation of Bitcoin's price holds significant global economic implications.

However, despite the growing body of literature, significant research gaps persist in the field of cryptocurrency price prediction. The cryptocurrency market is still relatively young and highly volatile. Therefore, achieving reliable predictive performance based solely on historical price data and technical indicators, as is common with traditional assets, remains challenging. According to previous research, unlike stock investors who can rely on a wide array of data to assess a company's value, cryptocurrency investors are highly responsive to news and social media because of the scarcity of established valuation metrics (*Kraaijeveld & De Smedt, 2018*; *Atashian & Khachatryan, 2018*). In this perspective, numerous studies have explored the correlation between public sentiment as reflected in textual data and Bitcoin price movements. However, most of these studies exclusively employed sentiment indices and price data (*Pano & Kashef, 2020*; *Georgoula et al., 2015*).

This creates a critical research gap in integrating more diverse and sophisticated data sources and methodologies to enhance predictive accuracy. Current approaches often overlook the potential of advanced machine learning models to incorporate various types of data, such as macroeconomic indicators, regulatory news, and correlations with traditional financial asset, which could significantly influence Bitcoin prices. In other words, there is a lack of comprehensive frameworks that systematically combine sentiment analysis with technical and macroeconomic indicators to provide a more holistic and robust prediction model. Addressing these gaps is essential for developing more accurate and reliable predictive models that can better serve investors and policymakers in navigating the complex and volatile environment of the cryptocurrency market.

In this research, the authors suggest a robust initiative aimed at predicting Bitcoin prices across diverse timeframes, including short-term, mid-term, and long-term intervals, harnessing the capabilities of time series analysis. The inputs used in this study extend beyond historical Bitcoin price data by incorporating an extensive range of features. These include not only the raw price but also a rich set of derived technical indicators, a sentiment

index sourced from Bitcoin-related social media platforms, news sources, and insights extracted from Google Trends. Additionally, the authors incorporated a comprehensive set of macroeconomic indicators, on-chain data revealing transactional details within the Bitcoin network, and data from traditional financial assets into the proposed prediction framework. This study employed a total of nine validation algorithms ranging from traditional statistical models to state-of-the-art (SOTA) machine learning models: support vector regressor (SVR), Extreme Gradient Boosting (XGBoost) Regressor, convolutional neural networks (CNN), recurrent neural networks (RNN), long short-term memory (LSTM), bidirectional-LSTM (Bi-LSTM), gated recurrent unit (GRU), iTransformer, and TSMixer. The performance metric utilized for the evaluation were root mean squared error (RMSE), mean absolute error (MAE), mean absolute percentage error (MAPE), and R-squared value. Particularly, Bi-LSTM with a window size of 3, outperformed SOTA models such as iTransformer and TSMixer in predicting Bitcoin prices, consistent with the volatility characteristics of Bitcoin. Although SOTA models excel at identifying recurring patterns such as periodicity, they proved inadequate for predicting the highly volatile behavior of Bitcoin.

Additionally, to determine the significance of the input features, the authors analyzed the gradients to identify which variables had the most impact on the prediction outcomes. Subsequently, an ablation test was performed to validate feature effectiveness, revealing that using sentiment, technical, on-chain data, traditional assets data, and macroeconomic data together achieves better performance than using price data alone.

In summary, the main goal of this research was to evaluate the predictive capability of various time series analysis frameworks and explore the optimal window size by considering a wide range of input features, with the aim of improving the accuracy of Bitcoin price predictions. Furthermore, the experimental findings of this study can be utilized when attempting to predict highly volatile assets. The main contributions can be summarized as follows:

- This study integrated a wide range of features beyond historical price data, including technical indicators, sentiment indices, Google Trends insights, macroeconomic indicators, on-chain data, and traditional financial assets.
- This study evaluated nine algorithms (SVR, XGBoost Regressor, CNN, RNN, LSTM, Bi-LSTM, GRU, iTransformer, TSMixer) for predicting Bitcoin prices, finding that Bi-LSTM with a window size of 3 outperforms SOTA models.
- The authors aimed for explainable artificial intelligence through feature importance analysis of 54 individual input features. Additionally, the ablation test results demonstrated that utilizing each proposed input feature in this study improves predictive performance compared to using price data alone.

The remainder of this article is structured as follows. Section 'Related Work' reviews the related work in the field of Bitcoin and time series analysis. Section 'Materials & Methods' describes the materials and methods used in this study, including the data sources, feature engineering and preprocessing, and the time series models employed. Section 'Results' presents the results of the experiments, highlighting the performance of

different models and input feature sets. Section 'Discussion' provides a discussion of the results and implications of this study. Finally, 'Conclusion' concludes the article with a summary of our findings, limitations and suggestions for future research.

## RELATED WORK

In this section, the authors first explore the factors that could potentially influence the price of Bitcoin. Subsequently, the authors discuss studies aimed at predicting Bitcoin prices using sentiment analysis. Finally, the authors examine studies that aimed to predict asset prices through time series analysis.

### Factors that could potentially influence the price of Bitcoin

With the rapid growth of the cryptocurrency market, numerous research into Bitcoin have been conducted. Owing to Bitcoin's limited history and considerable price volatility, researchers have focused on the elements that influence its price.

As reported by *Panagiotidis, Stengos & Vravosinos (2019)*, external influences such as interest rates and fluctuations in exchange rates have been observed to impact Bitcoin prices. Conversely, *Dyhrberg (2016)* found that the price of Bitcoin displayed little or even a negative correlation with various financial assets such as gold, the United States (US) dollar, and major stock market indices. *Ciaian, Rajcaniova & Kancs (2016)* revealed that speculative sentiment among investors substantially influences short-term price changes in Bitcoin.

In addition, *Kristoufek (2015)* postulated that fundamental factors such as real-world use in transactions and money supply play a role in shaping Bitcoin's long-term price trends. Investors' growing interest in cryptocurrencies intensifies this effect. Furthermore, the study detailed in *Hakimdas Neves (2020)* used Google search term queries related to Bitcoin and found that global interest in Bitcoin often precedes price surges. Conversely, prices tend to decline when concerns about a market collapse intensify.

Moreover, *Kaya (2018)* established a strong correlation between public interest and Bitcoin prices by employing Pearson's correlation coefficient to illustrate that Twitter sentiments can assist in predicting changes in Bitcoin prices (*Li et al., 2019*). Similarly, *Kraaijeveld & De Smedt (2020)* found that both Twitter sentiments and the volume of textual information can influence Bitcoin prices.

*Lee, Kim & Park (2019)* quantified the tone of news articles after Financial Monetary Committee meetings and measured monetary policy shocks. The monetary shock identified by the Vector Autoregressive model is linked to changes in short-term interest rates and has been confirmed to affect asset prices. Ultimately, this methodology could serve as a supplement to extracting market expectations regarding monetary policies. *Ma et al. (2022)* also conducted a shock response analysis following Federal Open Market Committee (FOMC) meetings to examine the impact of US monetary policy shocks on Bitcoin prices. The analysis revealed that in the days following the FOMC meetings, the Bitcoin price decline had an observed effect. Furthermore, quantile regression confirms that monetary policy shocks have a greater impact on Bitcoin prices during periods of market prosperity. *Pyo & Lee (2020)* examined the impact of FOMC meetings and macroeconomic

**Table 1  Summary of studies on factors that could potentially influence Bitcoin price.**

| References | Selected factors |
| --- | --- |
| *Panagiotidis, Stengos & Vravosinos (2019)* | Interest rates, exchange rate |
| *Dyhrberg (2016)* | Other types of financial assets, such as gold, dollars, and major stock market indices |
| *Ciaian, Rajcaniova & Kancs (2016)* | Speculative sentiment of investors |
| *Kristoufek (2015)* | Real use in transactions, money supply, and investors' interest |
| *Hakimdas Neves (2020)* | Global interest, fear of market collapse |
| *Kaya (2018)* | Public interest |
| *Li et al. (2019)* | Twitter sentiment |
| *Kraaijeveld & De Smedt (2020)* | Twitter sentiment and the text volume |
| *Lee, Kim & Park (2019)* | Tone of news articles after the meetings of the Financial Monetary Committee |
| *Ma et al. (2022)* | FOMC meetings (US monetary policy shocks) |
| *Pyo & Lee (2020)* | FOMC meetings and macroeconomic announcements |
| *Mizdrakovic et al. (2024)* | EUR/USD exchange rates, Ethereum closing prices, GBP/USD exchange rates |
| *Todorovic et al. (2023)* | Past trends, asset performance, economic indicators, news articles, historical price and volume data, external factors |
| *Strumberger et al. (2023)* | Closing prices of Bitcoin, Ethereum, daily volume of Bitcoin-related tweets |

announcements on Bitcoin prices. The authors found that these events had different effects before and after FOMC meetings.

*Mizdrakovic et al. (2024)* introduced a two-layer framework for accurately predicting Bitcoin prices. The authors utilized data including Bitcoin, Ethereum, S&P 500, and Chicago Board Options Exchange Volatility Index (VIX) closing prices, exchange rates of the Euro (EUR) and Great British Pound (GBP) to United States dollar (USD), and the number of Bitcoin-related tweets per day. Consequently, the authors confirmed that EUR/USD exchange rates, Ethereum closing prices, and GBP/USD exchange rates had a significant impact on the predictions. *Todorovic et al. (2023)* aimed to predict future trends in Bitcoin prices by investigating various variables. To achieve this, the authors used data such as past trends, asset performance, economic indicators, news articles, historical price and volume data, and external factors. In particular, the authors demonstrated excellent performance of the Hybrid Adaptive Reptile Search Algorithm (HARSA)-LSTM model. *Strumberger et al. (2023)* predicted Bitcoin closing prices using daily trading volumes of Bitcoin and Ethereum closing prices, along with Bitcoin-related tweet counts. The authors demonstrated the superiority of the Bi-LSTM-HARSA methodology through quantitative analysis. The factors that have the potential to impact Bitcoin prices are summarized in Table 1.

Additionally, various models have been proposed to understand the movements of cryptocurrencies. *Gupta & Nalavade (2023)* aimed to accurately predict Bitcoin price values using extracted features and original features with a two-stage ensemble classifier. *Behera, Nayak & Kumar (2023)* simulated and predicted the behavior of four rapidly growing

cryptocurrencies including Bitcoin, Litecoin, Ethereum, and Ripple using a hybrid model. The authors proposed that the Chemical Reaction Optimization-Artificial Neural Network model could be utilized for price prediction. *Salb et al. (2022)* applied the enhanced sine cosine methodology to Support Vector Machine for hyperparameter tuning and attempted predictions for various cryptocurrencies including Bitcoin.

However, as the cryptocurrency market and ecosystem continue to mature, the influencing factors are constantly evolving, making predicting Bitcoin's prices a complex challenge when only these fragmented factors are considered. Therefore, further research is necessary to understand the factors that affect Bitcoin price formation.

## Research on Bitcoin price using sentiment analysis

Natural language processing (NLP) techniques are used widely in various domains to extract valuable insights from textual data for practical applications. This section reviews the relevant literature that employs NLP methods to gain insights into Bitcoin. Recent research has focused on utilizing social media, online forums, and news posts to gauge investor and public sentiments to predict Bitcoin price changes. *Ortu et al. (2022)* examined the correlation between discussions on social media and fluctuations in cryptocurrency market prices. The authors used statistical and NLP models, specifically Dirichlet multinomial regression (DMR). According to *Pano & Kashef (2020)*, the sentiment scores obtained through the Valence Aware Dictionary and Sentiment Reasoner (VADER) during the COVID-19 pandemic exhibited a significant correlation with short-term shifts in Bitcoin prices. *Georgoula et al. (2015)* employed support vector machines (SVM), regression models, and Twitter sentiment to evaluate changes in Bitcoin prices. *Lamon, Nielsen & Redondo (2017)* used a logistic regression approach to analyze tweets and news headlines to predict price fluctuations in Bitcoin and Ethereum, achieving an accuracy of 43.9% for price increases and 61.9% for price decreases.

In *McMillan et al. (2022)*, NLP algorithms were utilized to assess the relationship between investment sentiment and Bitcoin price fluctuations by analyzing data from the subreddits "r/bitcoin" and "r/investing." *Critien, Gatt & Ellul (2022)* predicted the direction and magnitude of Bitcoin price changes by employing sentiment analysis and post-volume data from Twitter. A model based on RNN and CNN achieved an accuracy rate of 63%. *Sattarov et al. (2020)* affirmed that sentiment analysis of Bitcoin-related tweets could be used to predict changes in Bitcoin prices. The authors achieved an accuracy rate of 62.48% using a random forest regression model for applying sentiment analysis to Twitter data.

In summary, building on insights from past research, it is evident that utilizing social media or news data offers a valid avenue for assessing overall investor sentiment. Therefore, the author emphasizes the potential to enhance prediction accuracy by integrating technical indicators, relationships with traditional assets, macroeconomic factors, and blockchain on-chain data into the prediction process.

## Research on predicting asset prices through time series analysis

Time series analysis is a statistical methodology applied to regularly collected data intervals and presents sequential arrangements over time (*Box et al., 2015*). This approach enables

the comprehension and prediction of patterns, trends, seasonality, and other data traits (*Hamilton, 2020*). Its application spans diverse fields, such as stock prices, economic indicators, weather, and stock market trends, for analysis and prediction.

*Moghar & Hamiche (2020)* applied an RNN to predict forthcoming stock market values, specifically focusing on LSTM. *Eapen, Bein & Verma (2019)* described a novel deep learning model that combines a CNN and LSTM to minimize overfitting effects while enhancing the predictive accuracy of the S&P 500. *Parekh et al. (2022)* attempted to predict the prices of various altcoins by considering the interdependence among Bitcoin, other cryptocurrencies, and market sentiment using a combination of GRU and LSTM. *Livieris, Pintelas & Pintelas (2020)* introduced a model that utilizes convolutional layers to extract valuable insights and comprehend the internal structure of time series data for gold price prediction. The authors also highlighted the efficacy of the LSTM layers in recognizing both short- and long-term relationships. *Cao et al. (2020)* introduced a new approach called DC-LSTM to capture complex couplings for predicting the USD/CNY exchange rates. This method establishes a deep structure composed of stacked LSTM to model intricate couplings. *Busari & Lim (2021)* proposed a hybrid model combining GRU with the adaptive boosting (AdaBoost) algorithm and compared its predictive performance on crude oil prices with that of the existing AdaBoost-LSTM model, demonstrating superior performance. *Rajabi, Roozkhosh & Farimani (2022)* introduced the concept of learnable window size and attempted to predict Bitcoin prices. The authors predicted Bitcoin prices using 42 input features, including on-chain data. However, no attempt has been made to merge diverse datasets from various input categories to predict Bitcoin prices.

# MATERIALS & METHODS

This section describes the overall experimental flow (Fig. 1). The point to note here is that an earlier version of this article is based on the author's doctoral dissertation entitled "Analyzing Bitcoin Perception and Predicting Prices with Sentiment Index, Technical Indicators, and External Factors."

## Data collection

The present study collected data in five large categories for 2,557 days, encompassing the timeframe from March 1, 2017, to March 1, 2023. The first dataset comprises historical Bitcoin price information in US dollars, including daily opening, closing, lowing, volume, fluctuation, and date. The first data were collected in comma separated values (CSV) file format by setting the desired date range and downloading it from investing.com. The second dataset comprised text data and numerical data employed to obtain sentiment information for Bitcoin. The text dataset includes news data collected from LexisNexis and social media data collected from Reddit (*Lee et al., 2023*; *Jung et al., 2024*). The LexisNexis data was gathered by accessing Lexis+ and using the queries "Bitcoin" and "BTC." A Python crawler was employed to collect the data in DOCX format, which was then preprocessed and merged into CSV file format. Access to the LexisNexis database may require an institutional paid subscription. The Reddit data was downloaded directly from the Reddit data archive, Pushshift, and parsed using the keyword "Bitcoin," then integrated into single

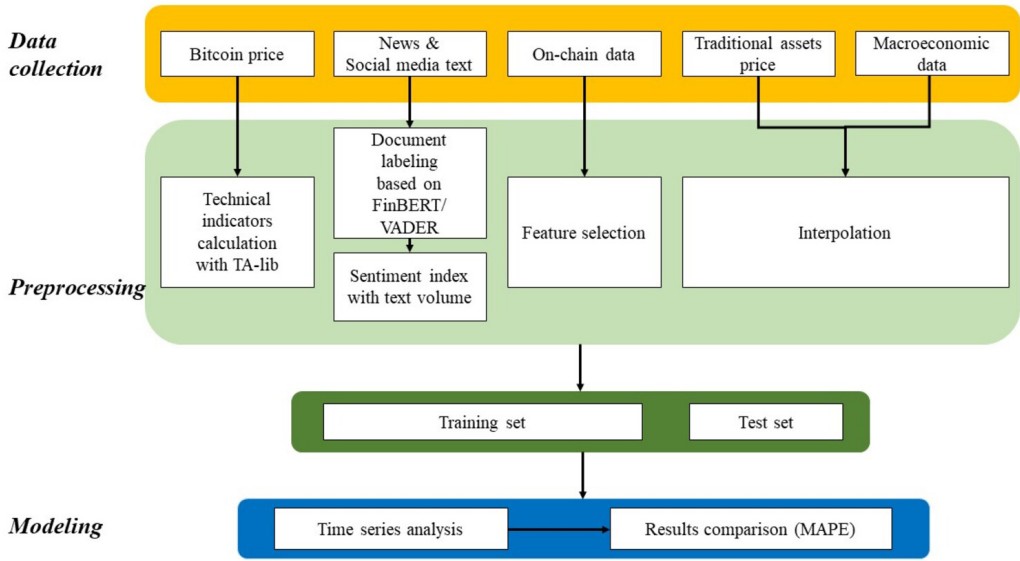

**Figure 1  Experimental flowchart for predicting Bitcoin price.**

CSV file format. For the numerical data used to assess sentiment, the authors collected data from Google Trends and the daily number of tweets. The Google Trends data were gathered by querying "Bitcoin" on Google and downloading the results. The tweet volume was collected using a Python scraper from bitinfocharts.com in CSV format. The third dataset was Bitcoin's on-chain data. On-chain data refers to information associated with the actual transaction data on the blockchain, covering all transactions and their detailed information on the Bitcoin network. The authors collected 12 Bitcoin on-chain data from bitinfocharts.com using a Python scraper in CSV format. The fourth dataset included information related to traditional assets. This data was collected from Investing.com in CSV format by downloading it to analyze its correlation with Bitcoin prices, including the S&P 500, the Dollar Index, gold price, and the US 10-year Treasury yield (Fig. 2). The fifth dataset employed was macroeconomic data collected to consider macroeconomic factors. These data included the US consumer price index (CPI), producer price index (PPI), and nonfarm employment payrolls and collected from investing.com (Table 2).

## Feature engineering and preprocessing

This section explores how data is preprocessed and transformed into features for use within the framework. This process includes the calculation of technical indicators and sentiment indices, the selection of applicable on-chain data, and the interpolation of traditional assets and macroeconomic indicators. Table 3 shows example of data used for feature engineering.

### Technical indicators

Technical indicators commonly employed to predict asset prices were chosen based on previous studies (*Oriani & Coelho, 2016*; *Shynkevich et al., 2017*; *Zhai, Hsu & Halgamuge,*

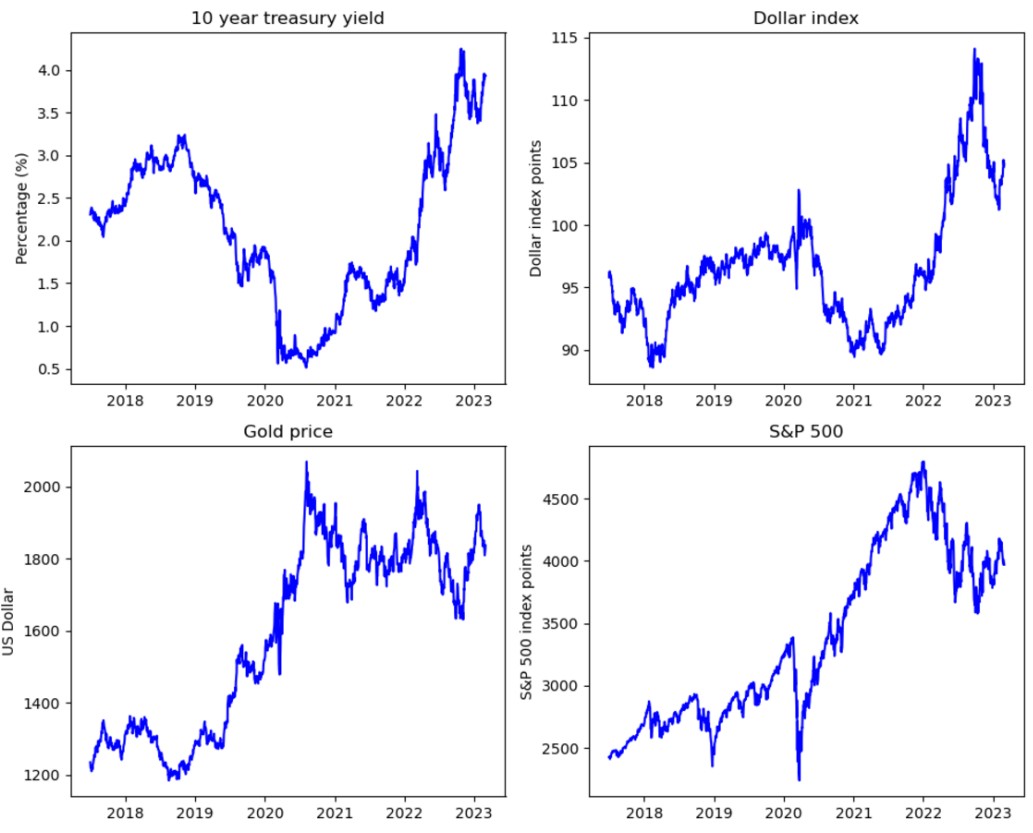

**Figure 2  Visualization of collected traditional assets data.**

**Table 2  Description of collected macroeconomic data.**

| Variable | Meaning by variable |
| --- | --- |
| PPI from the previous year & month | Index that measures the price fluctuations of raw materials, machinery and equipment, labor, and other expenses paid by companies producing goods and services. PPI is used to measure changes in producer prices over time, thereby assessing inflation in the economy. |
| CPI from the previous year & month | Index that measures the price fluctuations of goods and services purchased by consumers. Influencing government policy-making and monetary policy decisions, CPI serves as an evaluation tool for economic health and price stability. |
| Nonfarm employment index | Nonfarm Payrolls are one of the indicators measuring employment trends and are used to understand economic health. |

*2007*; *Rosillo, De la Fuente & Brugos, 2013*; *Ellis & Parbery, 2005*; *De Souza et al., 2018*; *Wang & Kim, 2018*; *Ni, Liao & Huang, 2015*). The Python technical analysis library (ta library) was used to obtain the selected technical indicators from the historical Bitcoin price. When calculating technical indicators such as moving averages, null values appear at the beginning of the data for the period equivalent to the moving average being used.

**Table 3  Example of input data after preprocessing (retaining only representative inputs for clarity).**

|  | Date | 2017-07-01 | 2017-07-02 |
|---|---|---|---|
| Bitcoin historical data | Close | 2,424.6 | 2,536.5 |
|  | Open | 2,480.6 | 2,424.6 |
|  | High | 2,529.6 | 2,555.3 |
|  | Low | 2,387.5 | 2,375.9 |
|  | Trading_Volume | 66,320 | 67,760 |
| Technical indicators | RSI_3 | 17.7642 | 57.8731 |
|  | SMA_3 | 2,487.8667 | 2,480.5667 |
|  | EMA_3 | 2,472.8591 | 2,504.6795 |
|  | MACD | 15.6684 | 13.3895 |
|  | signal | 57.3064 | 48.5229 |
|  | Stochastic RSI_fastk | 0.0 | 80.1804 |
|  | Stochastic RSI_fastd | 33.0489 | 33.0725 |
|  | Stochastic Oscillator Index_slowk | 54.6439 | 53.2359 |
|  | Stochastic Oscillator Index_slowd | 60.7049 | 56.9120 |
|  | WilliamR | $-76.7582$ | $-54.1430$ |
|  | Momentum | 12.0 | 43.8999 |
|  | ROC | $-9.4487$ | $-6.8422$ |
|  | BBands_upper | 2,915.5746 | 2,914.5674 |
|  | BBands_lower | 2,349.8121 | 2,353.7459 |
| On-chain data | mining_difficulty | 711,697,198,174.0 | 708,707,940,676.0 |
|  | transaction_value | 25,549.0 | 22,251.0 |
|  | bitcoin_sent | 3,510,026,648.0 | 2,919,890,153.0 |
|  | hash_rate | 5.7662e+18 | 6.2663e+18 |
|  | unique addresses | 553,869.0 | 522,589.0 |
| Sentiment indicators | sent_index_Reddit | $-0.1068$ | $-0.1133$ |
|  | post_volume_Reddit | 4,933 | 4,049 |
|  | post_volume_Lexis | 4 | 3 |
|  | sent_index_Lexis | $-0.75$ | $-0.3333$ |
|  | Google_trends | 13.0 | 13.2258 |
|  | Bitcoin Tweets | 42,000.0 | 41,914.4545 |
| Traditional assets | 10_year_treasury | 2.3040 | 2.3260 |
|  | Dollar_index | 95.8267 | 96.0233 |
|  | S&P_500 | 2,425.2766 | 2,427.1433 |
|  | Gold_price | 1,234.6 | 1,226.9 |
| Macroeconomic indicators | CPI_year | 1.878571 | 1.857143 |
|  | CPI_month | 0.1 | 0.1 |
|  | PPI_year | 2.084615 | 2.069231 |
|  | PPI_month | 0.002846 | 0.002692 |
|  | Nonfarm payment | 150,000.0 | 162,000.0 |

Consequently, data before July 1, 2017, were excluded from the experiment. A description of the technical indicators used in the experiment is shown in Table 4, and the representative technical indicators employed in the experiment are shown in Fig. 3.

**Table 4 Description of utilized technical indicators.**

| Variable | Meaning by variable |
|---|---|
| SMA (simple moving average) | Calculation of the average price of an asset over a specific timeframe. |
| EMA (exponential moving average) | An indicator that can be utilized to assess short-term trends by assigning greater weight to recent data points while diminishing the significance of historical values. |
| RSI (relative strength index) | The RSI ranges from 0 to 100 for a specific asset. When it approaches 100, it is considered over-bought, and when it approaches 0, it is considered oversold. |
| MACD (moving average convergence divergence) | MACD racks trends by illustrating the interplay between two moving averages, helping identify potential purchase or sale signals. |
| Stochastic RSI | Stochastic RSI, applying the stochastic oscillator to RSI, analyzes the upward and downward trends of an asset. |
| Stochastic oscillator | The stochastic oscillator reveals the current position of the asset price within the high and low price range of a specific period. |
| Momentum | Momentum measures the speed of price fluctuations. Strong momentum signifies dynamic price movements, while weak momentum means price stability. |
| Rate of change (ROC) | ROC measures the percentage change in price over a specified period, providing insights into how quickly the price of an asset is changing. |
| Bollinger bands | Bollinger Bands primarily aim to monitor price movements in relation to the middle band and predict future price volatility based on their proximity to the upper and lower bands. |

### Sentiment index

Duplicate and missing values were eliminated from the textual data obtained from LexisNexis and Reddit, and unnecessary words were removed. To process the Reddit data, VADER was used to preprocess the text and generate a compound score, as outlined by *Hutto & Gilbert (2014)*. A threshold of 0.1 was established for the compound score based on *Wu et al. (2022)*. For the LexisNexis dataset, a sentiment analysis was conducted using FinBERT to classify sentiments into positive, negative, and neutral labels (*Araci, 2019*; *Lee et al., 2024c*). FinBERT is a BERT model extensively trained on economic news, and is considered the most appropriate choice for sentiment analysis in this domain (*Lee et al., 2024a*). The sentiment index was then computed as the difference between the number of negative and positive texts for a specific date (*Wu et al., 2022*). The calculation of sentiment index is provided in Eq. (1).

$$\text{Sentiment index} = \frac{M_{tpos} - M_{tneg}}{\text{Total text}} \tag{1}$$

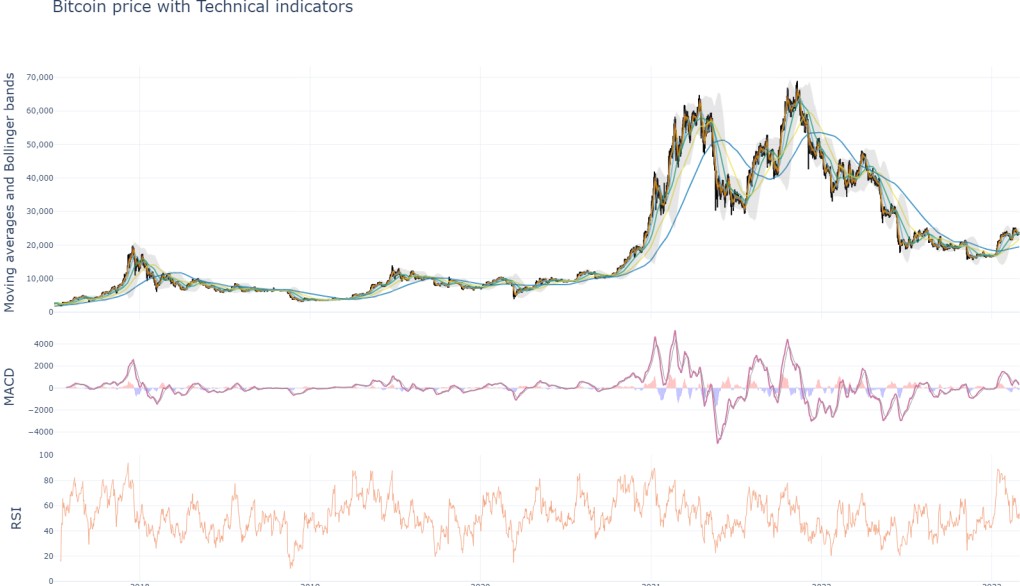

**Figure 3 Bitcoin price with representative technical indicators.**

**Table 5 Description of utilized data for assessing investor sentiment.**

| Variable | Meaning by Variable |
| --- | --- |
| Reddit sentiment by day | The sentiment indicators for each day obtained through calculations using the sentiment calculation equation on the results extracted using VADER. |
| Reddit volume by day | The number of Reddit posts (submission + comments) from the subreddit "Bitcoin" for each day. |
| Lexis sentiment by day | The sentiment index for each day obtained through the sentiment index calculation equation applied to the results extracted using FinBERT. |
| Lexis volume by day | The number of news articles on "Bitcoin" for each day. |
| Bitcoin tweets | The number of tweets on "Bitcoin" for each day. |
| Google trends | The daily search volume for the query 'Bitcoin' on Google. |

where $M_{tpos}$ is the total number of positive texts and $M_{tneg}$ is the total number of negative texts on day t. Consistent with previous studies that highlighted the impact of post-volume on price fluctuations (*Kraaijeveld & De Smedt, 2020*; *Critien, Gatt & Ellul, 2022*), the total volume of texts was incorporated. Figure 4 and Table 5 provide a comprehensive review of the processed data used to evaluate sentiments.

### On-chain data selection

To initially select optimal features for the prediction, the Pearson correlation coefficient between the Bitcoin closing price and each on-chain feature was analyzed (Fig. 5, Table 6). In Pearson's correlation analysis, a positive correlation coefficient between two variables

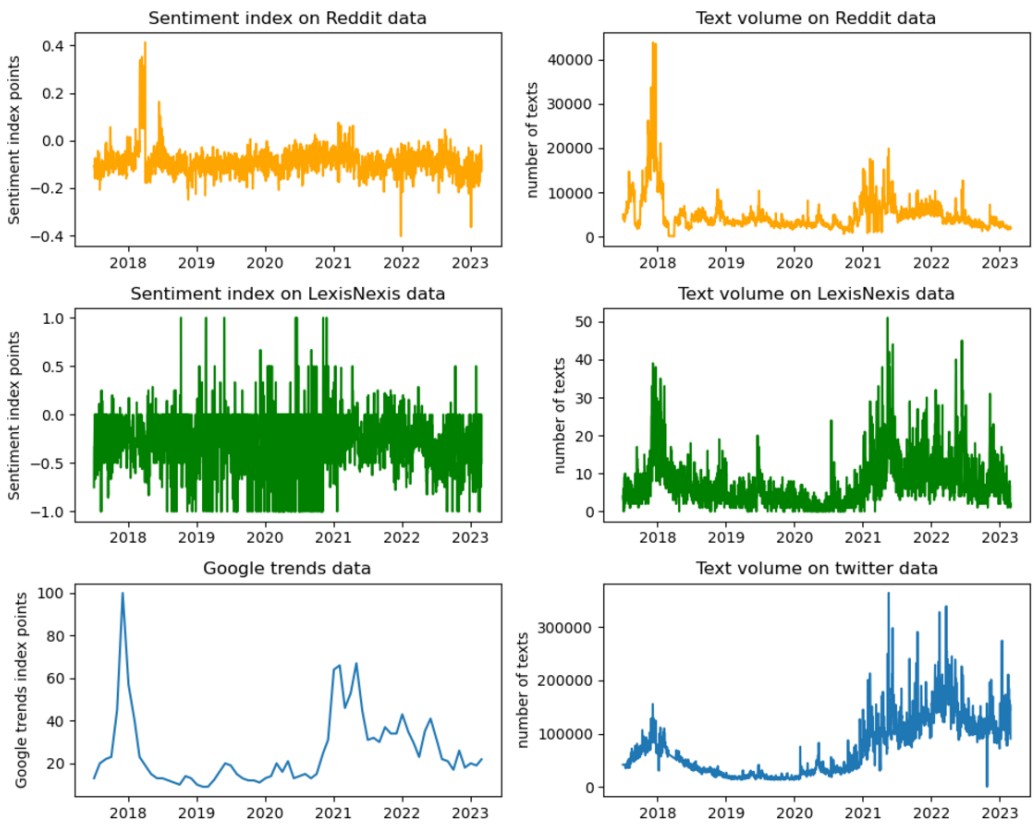

**Figure 4** Visualization of utilized data for assessing investor sentiment.

implies that when $x$ increases, $y$ increases, whereas a negative coefficient implies that when $x$ decreases, $y$ increases (*Pearson, 1895*). Variables with a correlation coefficient below 0.3 were removed under the assumption that they have a low correlation with the closing price of Bitcoin. Additionally, variables exceeding a correlation coefficient of 0.9 were also excluded. As a result, five indicators were used as inputs for the on-chain data: average mining difficulty, average transaction value, Bitcoin_sent, hash rate, and number of unique addresses (Fig. 6).

### Interpolation

As stated previously, traditional asset data encompass stocks, dollar index, and US 10-year Treasury yield. However, the cryptocurrency market's continuous operation, functioning 24/7 throughout the year, contrasts with traditional assets that observe non-trading weekends. This disparity leads to data deficiency during weekends, necessitating methods to bridge these gaps. Additionally, the numerical values of macroeconomic data are periodically released. Therefore, linear interpolation was performed due to the unavailability of daily data.

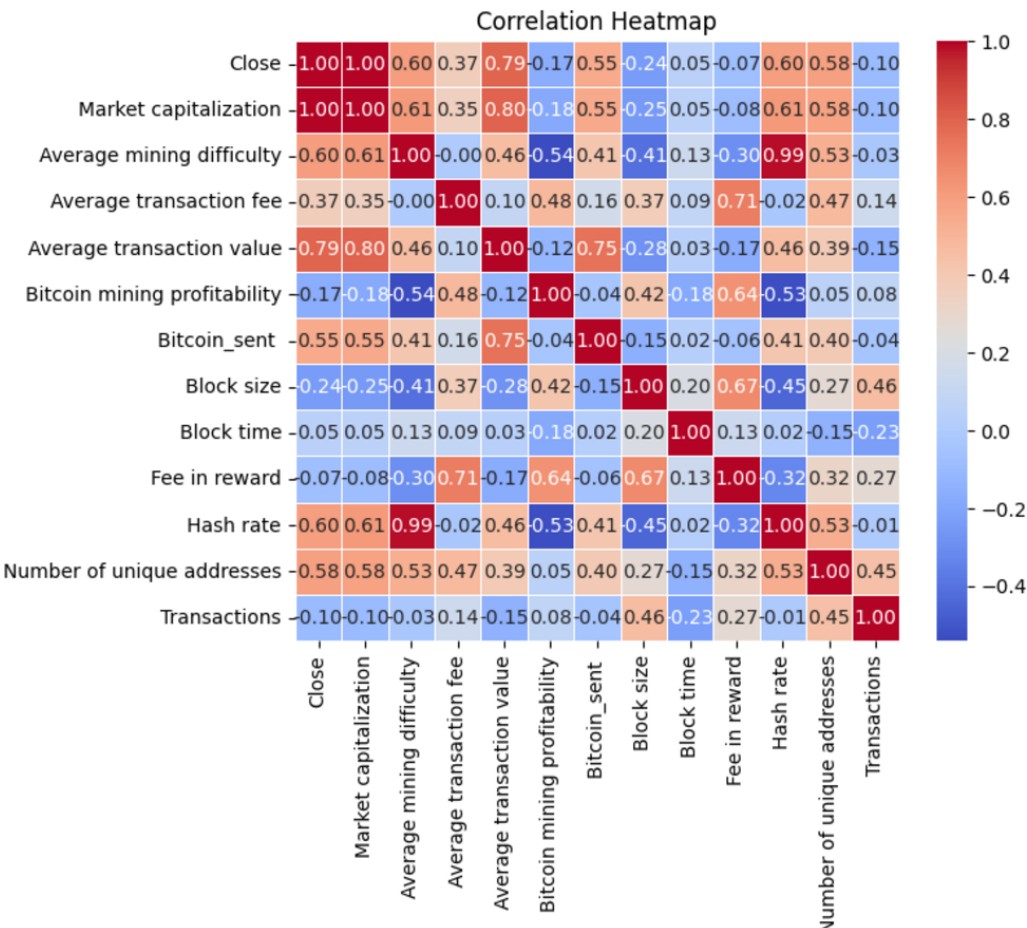

**Figure 5** Pearson correlation coefficient heatmap Bitcoin price and on-chain features.

## Consideration for the optimal window size

The window size in time series analysis refers to the number of past data points used by the model for making predictions. For instance, a window size of 3 means that the model uses data from the past 3 days to predict the next day (*Selvin et al., 2017*; *Rajabi, Roozkhosh & Farimani, 2022*). Experiments exploring the effectiveness of various input window sizes for the short-term (3 and 5 days), mid-term (14 and 30 days), and long-term (60 and 120 days) were executed to identify the most effective window size for predicting Bitcoin prices. Regardless of the input window size, the output window size was fixed for a single day (Fig. 7).

Considering the optimal window size is crucial for the following important reasons. First, in time series analysis, the window size directly impacts the model's ability to capture temporal dependencies and patterns. Different window sizes can significantly alter the model's performance by either incorporating too much noise or missing important long-term dependencies. Second, by examining the difference in window size, the authors can understand how sensitive the proposed model is to the length of input data. This is

**Table 6  Description of collected on-chain data.**

| Variable | Correlation coefficient | Meaning by variable |
|---|---|---|
| Market capitalization | 0.9991 | The total value of the Bitcoin market in US dollars. |
| Average mining difficulty | 0.5978 | As mining competition intensifies, the cryptographic problem's complexity for miners increases, and when competition eases, it reduces the block generation function's difficulty. |
| Average transaction fee | 0.3652 | Average transaction fee refers to the average fee paid by users to process Bitcoin transactions. It provides information about network congestion and the volatility of transaction fees. |
| Average transaction value | 0.7917 | Average transaction value calculates the average amount of Bitcoin sent or received in all transactions on the Bitcoin network. It provides statistical information about the level of network activity and the transaction amounts. |
| Bitcoin mining profitability | −0.1661 | Bitcoin mining profitability is an indicator that reflects how much profit Bitcoin miners can earn during the process of mining Bitcoin. |
| Bitcoin_sent | 0.5466 | Bitcoin_sent is an indicator that represents the total amount of Bitcoin sent over a specific period in the Bitcoin network. Bitcoin_sent assists in understanding active Bitcoin transaction activity or the transaction volume during specific periods. |
| Block size | −0.2405 | The total size of transactions accepted in each block. |
| Block time | 0.0479 | Block time represents the average time it takes for a new block to be generated in the Bitcoin blockchain, reflecting the operational speed and performance of the blockchain network. |
| Fee in reward | −0.0737 | "Fee in reward" indicates the Bitcoin fees integrated into miners' rewards upon uncovering a Bitcoin block. These fees act as encouragements for transaction processing within the Bitcoin network and form part of miners' compensation for incorporating and handling transactions in a block. |
| Hash rate | 0.5956 | It is a measure of the performance of a minor device. In other words, the hash rate indicates the rate at which a miner succeeds in solving the hash to receive revenue. |
| Number of unique addresses per day | 0.5848 | This variable represents the number of unique addresses used in the Bitcoin network on a specific date. Each Bitcoin transaction includes the addresses of the sender and receiver, and this metric is used to count the total number of unique addresses used in all transactions generated on a specific date. |
| Transactions | −0.0964 | Transactions refer to the total count of transactions recorded on the blockchain for a particular day. |

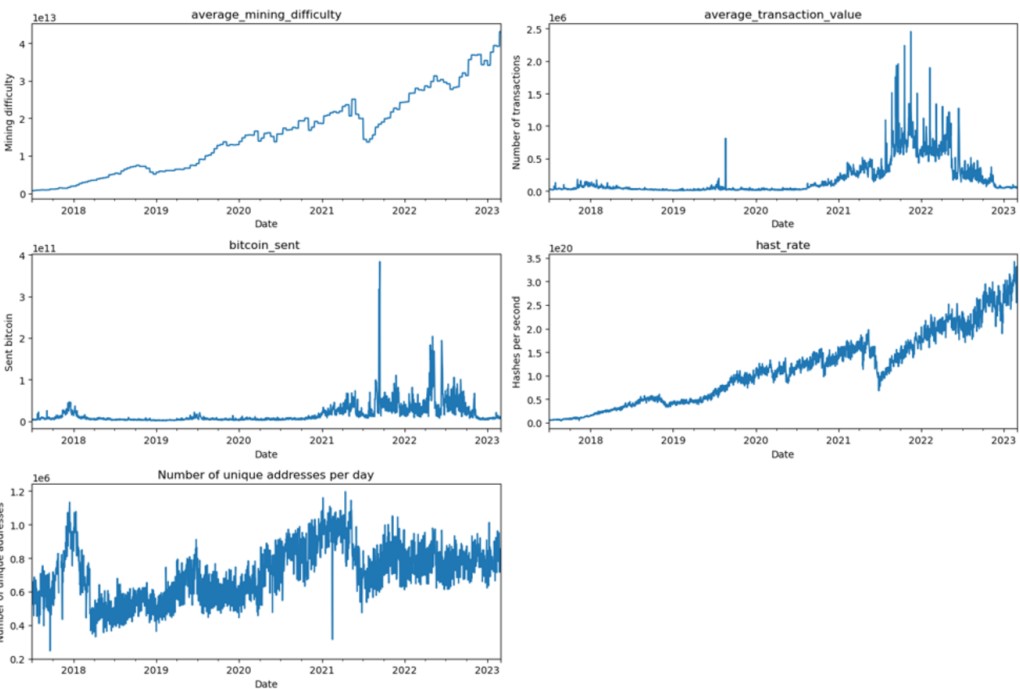

**Figure 6  Visualization of on-chain data selected for the experiment.**

crucial because the optimal window size can vary depending on the specific characteristics of the time series data, such as seasonality and trend components. Third, while increasing the number of epochs primarily addresses how well the model can learn from the training data, it doesn't directly address whether the model is receiving the most informative input data. Many epochs can lead to overfitting if the input data (*i.e.,* window size) is not optimal.

Finally, merged data were divided into a training test ratio of 80:20 for the experiment, and a min-max scaler was applied (*Lee, Kim & Jung, 2024b*; *Basnayake & Chandrasekara, 2024*; *Casado-Vara et al., 2021*; *Al-Alyan & Al-Ahmadi, 2020*).

## Time series analysis framework
### *Employed time series analysis algorithm*

This study employed nine experimental frameworks: SVR, XGBoost Regressor, CNN, RNN, LSTM, Bi-LSTM, GRU, iTransformer, and TSMixer. When considering each strength individually, SVR possesses the capability to perform well even with high-dimensional data, making it applicable to various types of datasets. On the other hand, the XGBoost Regressor enables efficient model training with large-scale datasets and is applicable to diverse types of data and problems. CNN excels in initial feature extraction by capturing local patterns in time series data. RNN is adept at reflecting temporal patterns in sequential data, while LSTM effectively tackles long-term dependency issues. Bi-LSTM improves predictive accuracy by leveraging past and future information in time series data, and GRU is noted for its computational efficiency. These models are extensively utilized in time series analysis and consistently deliver strong performance. Furthermore, iTransformer

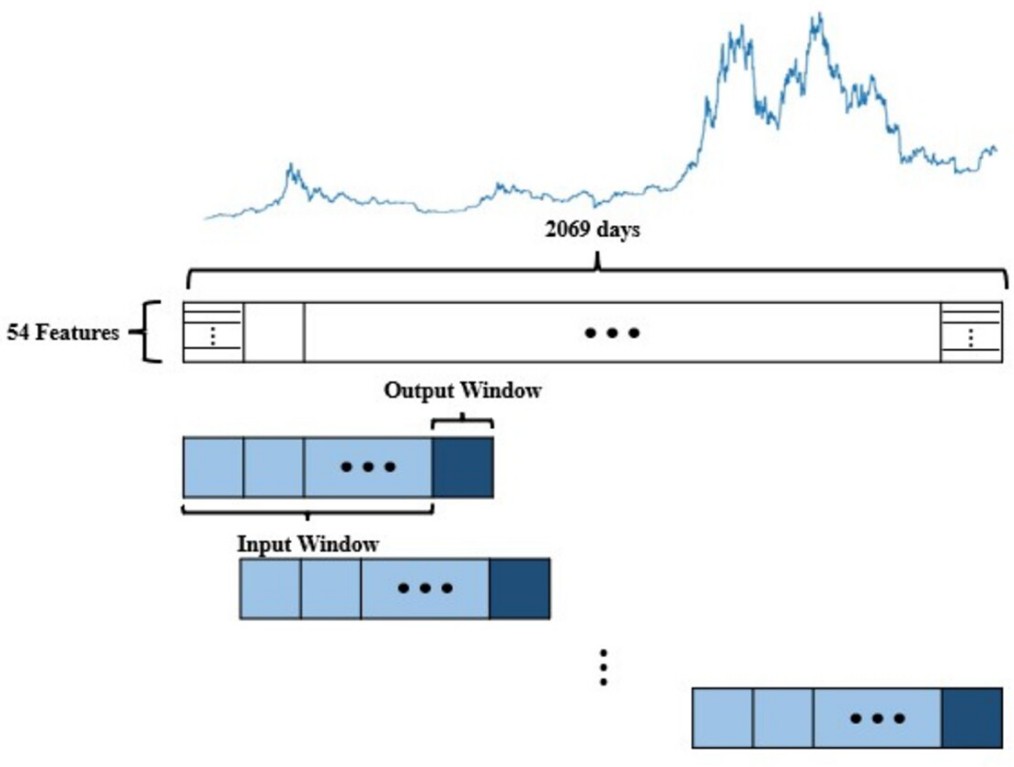

**Figure 7 Consideration for the optimal window size.**

and TSMixer are recognized as SOTA models in this field. By utilizing this wide range of models, the aim was to identify the most suitable model for processing features needed for Bitcoin price prediction. Brief descriptions of each framework are provided in this section.

SVR is a type of algorithm used for solving regression problems. Unlike SVM classifiers that find an optimal hyperplane to separate classes (*Boser, Guyon & Vapnik, 1992*), SVM regressors aim to find an optimal hyperplane that minimizes the error between predicted and actual values in a continuous output space. The main advantage of SVM regression lies in maximizing the margin, which enhances its generalization performance. This means the model operates robustly even with new data. SVM regression performs well even with large or complex datasets, although it is important to consider potential long training times.

XGBoost is a model that solves regression problems using the algorithm called XGBoost. XGBoost is an ensemble technique that combines multiple weak decision trees to form a powerful prediction model (*Chen & Guestrin, 2016*). Each decision tree is trained to complement the prediction errors of previous trees, ultimately providing high predictive accuracy. A significant advantage of XGBoost is its support for fast learning and classification speeds through parallel processing. Additionally, it retains the interpretability of tree-based models, allowing for the construction of models with strong explanatory power. However, a drawback is its susceptibility to overfitting.

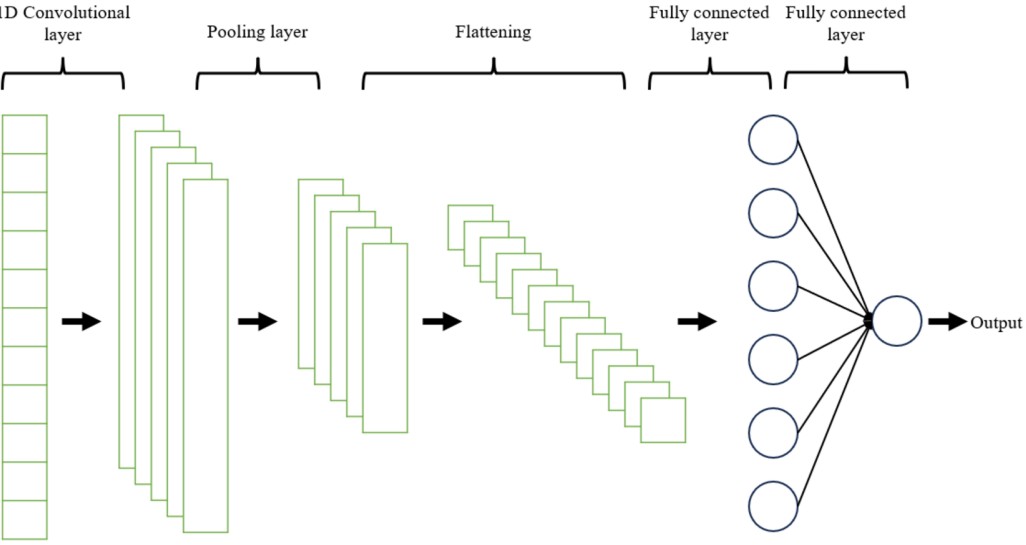

**Figure 8** **Basic CNN framework structure for time series prediction.** Source: compiled from *Zheng et al. (2014)*.

CNN is widely applied in image processing, but its value also extends to discovering and extracting patterns in sequential data (*Zheng et al., 2014*). By employing a series of filtering and pooling operations, CNN effectively detects features within the datasets. In the time series analysis, 1D convolutional layers are used instead of the 2D convolutional layers utilized in computer vision. The basic CNN framework structure for the time series prediction is shown in Fig. 8.

RNN is useful for processing sequential data by using the output from the previous step as the input for the current step (*Rumelhart, Hinton & Williams, 1986*). However, learning patterns from long sequences can be challenging owing to long-term dependencies. The internal cell structure of the RNN is shown in Fig. 9.

LSTM is a variation of an RNN that effectively learns long-term dependencies in sequence data while maintaining both short-term and long-term memories (*Hochreiter & Schmidhuber, 1997*). The LSTM uses three gates and cell states to preserve and regulate information, enabling an understanding of patterns over time. The internal cell structure of LSTM is illustrated in Fig. 10.

Bi-LSTM is employed to comprehensively grasp information by processing sequential data in both forward and backward directions (*Siami-Namini, Tavakoli & Namin, 2019*). The utilization of information from both directions in Bi-LSTM grants a higher capacity for expression than the unidirectional LSTM. This characteristic assists in gaining a deeper understanding of the complex patterns found in sequential data. The structure of Bi-LSTM is shown in Fig. 11.

The GRU is also a type of RNN designed to address the issue of long-term dependencies (*Cho et al., 2014*). The GRU employs reset and update gates. Through these two gates, the GRU effectively determines which information should be retained or discarded, optimizing

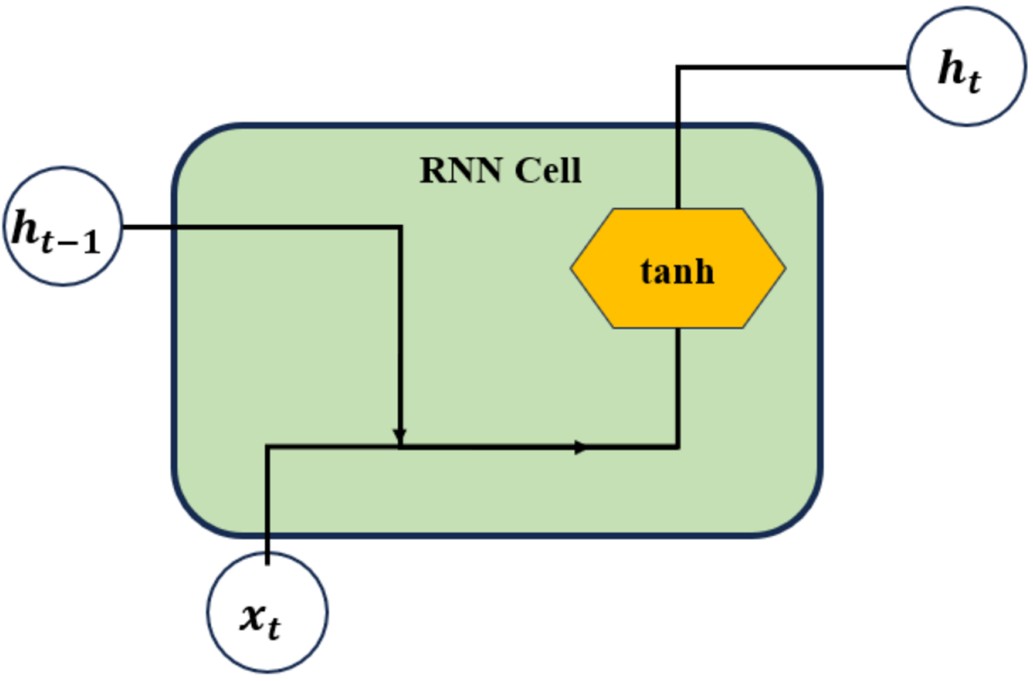

**Figure 9** **Internal cell structure of RNN.** Source: compiled from *Rumelhart, Hinton & Williams (1986)*.

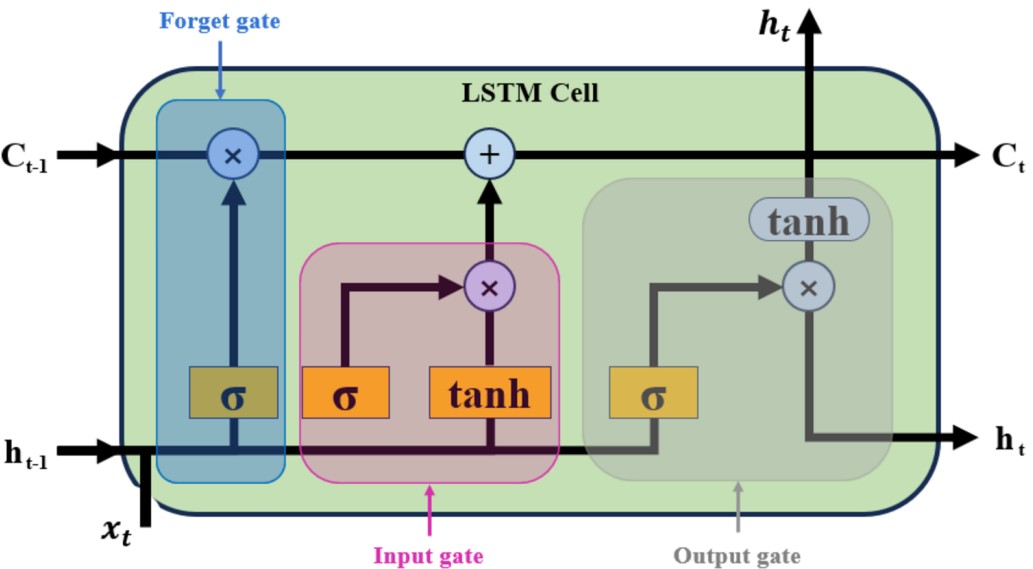

**Figure 10** **Internal cell structure of LSTM.** Source: compiled from *Hochreiter & Schmidhuber (1997)*.

the utilization of the information from the preceding steps. The internal cell structure of the GRU is illustrated in Fig. 12.

iTransformer is a novel adaptation of the Transformer architecture for time series forecasting without modifying its basic components (*Liu et al., 2023*). The iTransformer
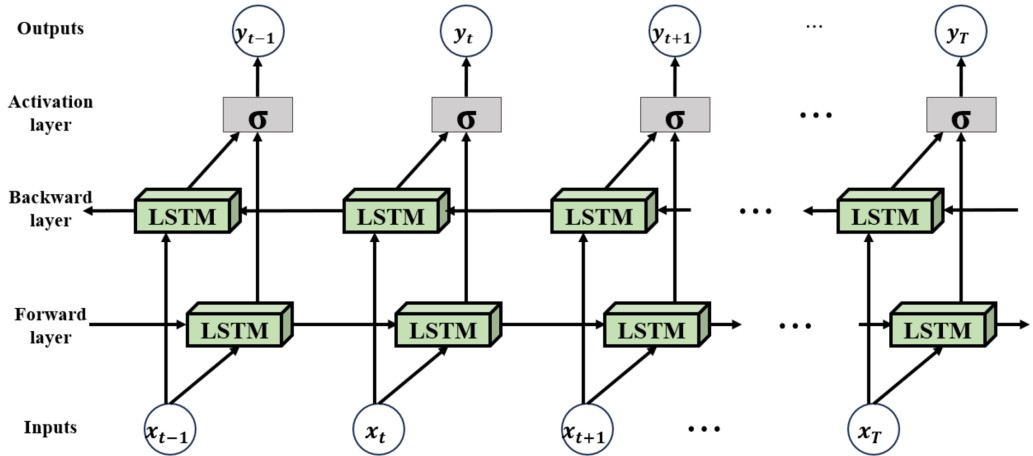

**Figure 11** **Bi-LSTM structure.** Source: compiled from *Siami-Namini, Tavakoli & Namin (2019)*.

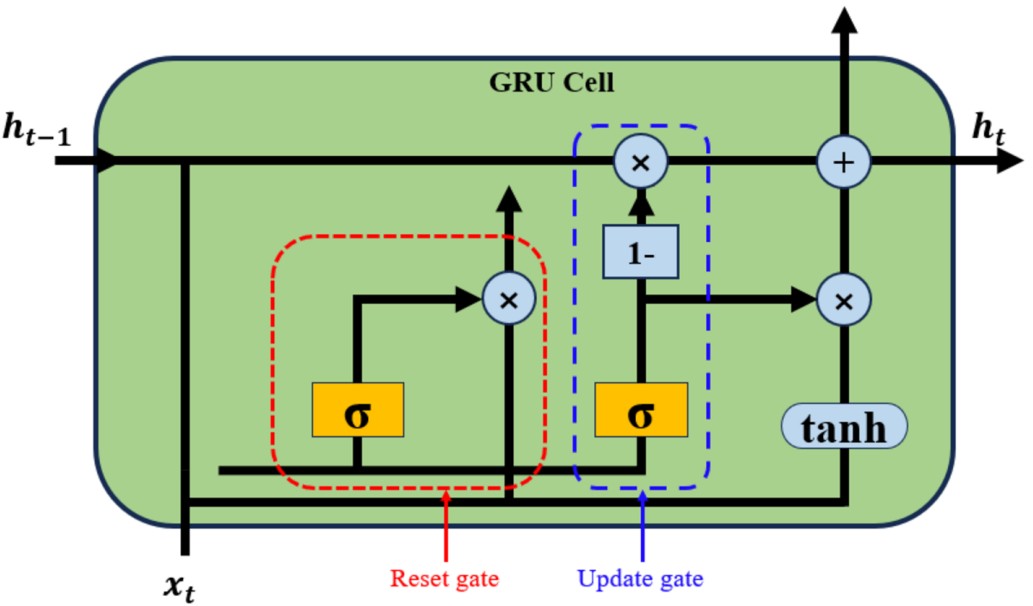

**Figure 12** **Internal cell structure of GRU.** Source: compiled from *Cho et al. (2014)*.

converts time into tokens representing different variables, enabling the capture of multivariate correlations through the attention mechanism. It employs a feed-forward network for each variable token to learn nonlinear representations. This model achieves SOTA performance on real-world datasets, enhancing the Transformer family with improved generalization across variables and better utilization of arbitrary lookback windows for forecasting time series. The overall structure of the iTransformer is illustrated in Fig. 13.

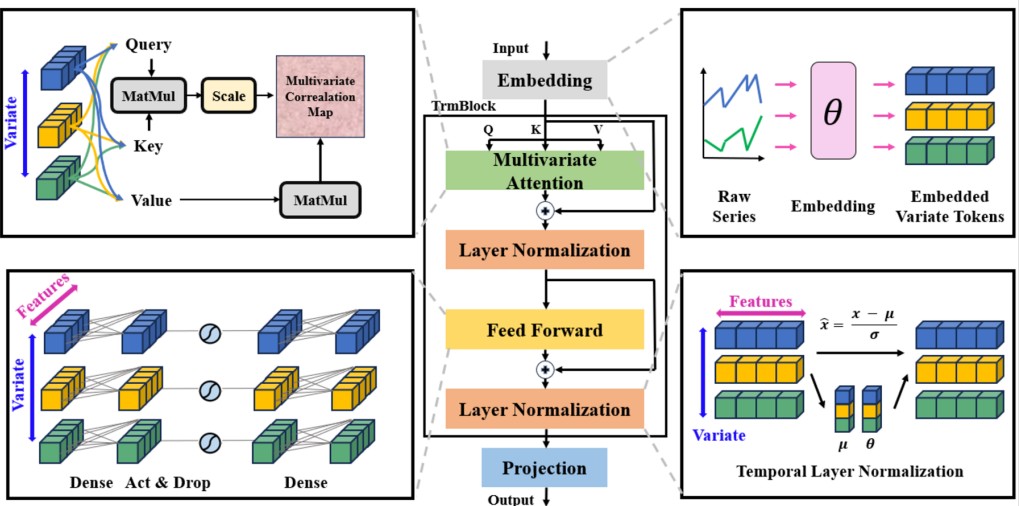

**Figure 13 ITransformer structure.** Source: compiled from *Liu et al. (2023)*.

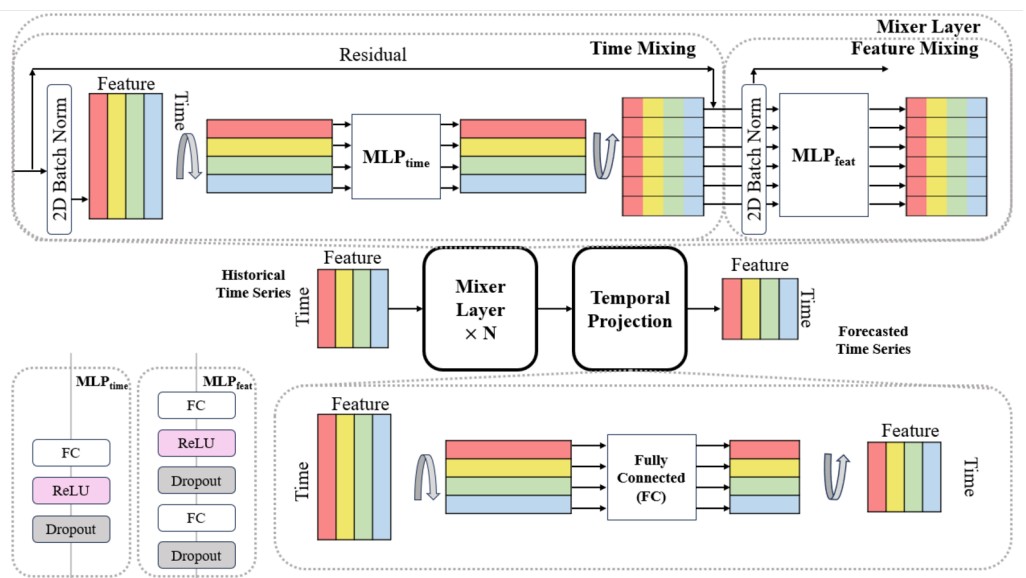

**Figure 14 Multivariate TSMixer structure.** Source: compiled from *Chen et al. (2023)*.

TSMixer is a novel architecture that stacks multi-layer perceptrons to efficiently extract information along both time and feature dimensions for capturing the complexity of multivariate time-series datasets (*Chen et al., 2023*). Despite its simplicity, the TSMixer performs comparably to specialized SOTA models on certain academic benchmarks and notably outperforms them on real-world retail data like the challenging M5 benchmark. The overall structure of the TSMixer is illustrated in Fig. 14.

*Experimental setup*

The authors utilized a computer with the following hardware configuration: NVIDIA GeForce RTX 3070 GPU, 32GB RAM, and Intel i7-10700F CPU. The experiments employed Python 3.10.12, and Python packages Torch 1.12.0, Tensorflow 2.15.0, and Sklearn 1.1.0. Learning rates varied as 0.0005, 0.001, 0.005, and 0.01, and batch sizes were tested at 8, 16, 32, and 64. Optimizers Adam and RMSprop were used, with epochs set to 10 and 20, and early stopping was applied with a patience of 2 epochs.

# RESULTS

The performances of the SVR, XGBoost regressor, CNN, RNN, LSTM, Bi-LSTM, GRU, iTransformer, and TSmisxer are experimentally evaluated in this section. Various window sizes were employed in the experiments, and an ablation test was conducted on the model that demonstrated the best performance.

## Performance evaluation

This study employed four metrics for model evaluation and their respective calculation can be found in Eqs. (2) to (5).

$$RMSE = \sqrt{\sum_{i=1}^{n} \frac{(y_i - \hat{y}_i)^2}{n}} \qquad (2)$$

$$MAE = \frac{1}{n} \times \sum_{i=1}^{n} |y_i - \hat{y}_i| \qquad (3)$$

$$MAPE = \frac{100}{n} \times \sum_{i=1}^{n} \left| \frac{y_i - \hat{y}_i}{y_i} \right| \qquad (4)$$

$$R\_squared = \frac{\sum_{i=1}^{n} (y_i - \hat{y}_i)^2}{\sum_{i=1}^{n} (y_i - \bar{y})^2}. \qquad (5)$$

In the equations above, $n$ represents the number of observations, $y_i$ denotes the actual observed values, $\hat{y}_i$ represents the predicted values, and $\bar{y}$ signifies the mean of the actual observed values, respectively.

These evaluation metrics offer comprehensive tools for assessing prediction accuracy across various domains such as finance, business, and supply chain management. MAPE, utilizing a ratio measurement that expresses prediction errors as percentages, highlights the relative disparity between predicted and actual values. This feature allows for effective alignment assessment with target values and facilitates straightforward comparisons among different time series data or prediction models. Its intuitive percentage representation makes it accessible even to non-experts, enhancing its utility in decision-making and evaluation.

In addition to MAPE, RMSE, R-squared value, and MAE further enhance the evaluation framework. RMSE quantifies the average magnitude of prediction errors, offering insights into the overall model fit to the data. R-squared value measures how well the model fits the data compared to a simple average of the target values, providing a measure of goodness of fit where higher values signify better alignment. Meanwhile, MAE provides a

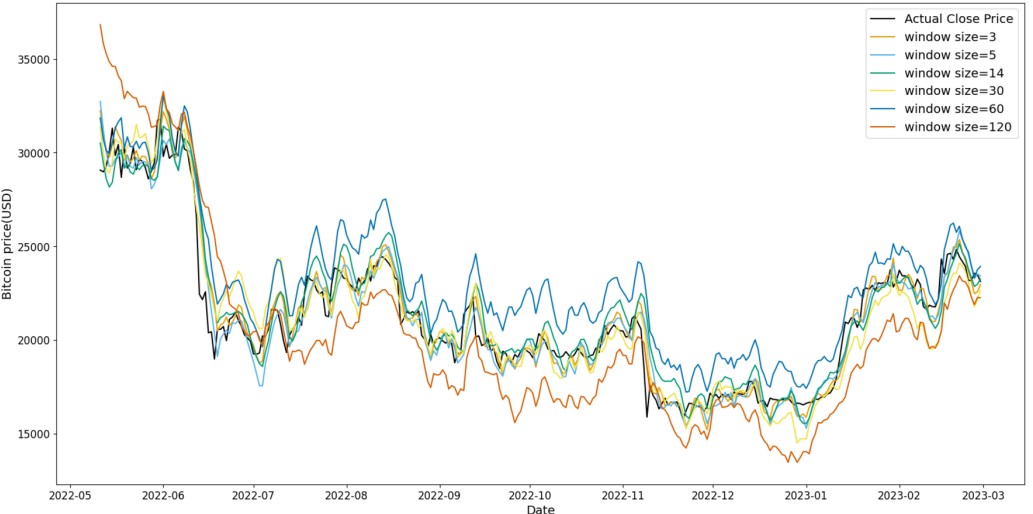

**Figure 15** **Visualization of predictions employing Bi-LSTM according to the different window sizes.**

straightforward measure of average error magnitude, supporting a clear understanding of prediction accuracy across different scenarios and models.

These metrics collectively function as valuable instruments for evaluating prediction accuracy, facilitating robust comparisons and informed decision-making across various contexts of predictive modeling.

## Experiment results based on the difference in window size

Because predictions for time series data with high volatility can occasionally be unstable, 30 trainings and predictions were conducted separately, and their averages using evaluation metrics were derived. A grid search was conducted to determine the optimal parameters, ultimately identifying the best outcome at a learning rate of 0.005, Adam optimizer, and a batch size of 64.

Overall, using a short-term window size showed better performance than employing a long-term window size. Ultimately, Bi-LSTM showed superior performance when utilizing a window size of 3, indicating that employing a short-term window size is notably more effective than using long-term window size predictions for highly volatile Bitcoin data (Table 7 and Fig. 15) (*Ji, Kim & Im, 2019*).

Specifically, the effectiveness of the 3-day window size can be attributed to its ability to capture short-term fluctuations and rapid changes inherent in Bitcoin prices. This granularity allows our model to react swiftly to market dynamics, thereby outperforming longer-term window predictions which may lag in responsiveness to volatile shifts. Additionally, the authors observed that shorter window sizes excel in capturing the immediate impact of market events and sentiment changes, crucial factors influencing Bitcoin's price movements. Conversely, longer-term windows smooth out these rapid changes, potentially leading to less accurate predictions during periods of high volatility.

**Table 7  Performance evaluation based on different window size.**

| | Metrics | Input window size | | | | | |
|---|---|---|---|---|---|---|---|
| | | 3 | 5 | 14 | 30 | 60 | 120 |
| SVM Regressor | RMSE | 0.03319 | 0.03526 | 0.04042 | 0.05868 | 0.09659 | 0.18238 |
| | MAE | 0.02869 | 0.03076 | 0.03566 | 0.05286 | 0.09053 | 0.17672 |
| | MAPE (%) | 9.47 | 10.27 | 12.11 | 18.15 | 30.81 | 62.19 |
| | R-squared | 0.94673 | 0.93928 | 0.91844 | 0.81903 | 0.41512 | −8.90925 |
| XGBOOST Regressor | RMSE | 0.03797 | 0.04713 | 0.07601 | 0.0887 | 0.10304 | 0.08795 |
| | MAE | 0.02822 | 0.03735 | 0.05859 | 0.07657 | 0.08143 | 0.13043 |
| | MAPE (%) | 7.23 | 10.99 | 14.23 | 21.41 | 22.86 | 29.03 |
| | R-squared | 0.93028 | 0.89151 | 0.71147 | 0.58652 | 0.33439 | −1.30425 |
| CNN | RMSE | 0.02473 | 0.02603 | 0.02994 | 0.03774 | 0.03749 | 0.07726 |
| | MAE | 0.02007 | 0.02083 | 0.02543 | 0.0328 | 0.03174 | 0.03038 |
| | MAPE (%) | 6.02 | 6.46 | 8.18 | 9.40 | 9.77 | 11.12 |
| | R-squared | 0.97042 | 0.96692 | 0.95524 | 0.92515 | 0.91191 | 0.6321 |
| RNN | RMSE | 0.02392 | 0.02645 | 0.02760 | 0.02873 | 0.03864 | 0.03936 |
| | MAE | 0.01835 | 0.02067 | 0.02217 | 0.02375 | 0.03061 | 0.03559 |
| | MAPE (%) | 5.16 | 5.85 | 6.67 | 7.54 | 9.75 | 12.55 |
| | R-squared | 0.97234 | 0.96584 | 0.96195 | 0.95661 | 0.90638 | 0.5386 |
| LSTM | RMSE | 0.01880 | 0.02182 | 0.02926 | 0.03382 | 0.03263 | 0.03446 |
| | MAE | 0.01350 | 0.0174 | 0.02316 | 0.02782 | 0.02822 | 0.03059 |
| | MAPE (%) | 3.67 | 5.09 | 6.69 | 8.25 | 8.85 | 10.85 |
| | R-squared | 0.98190 | 0.97676 | 0.95725 | 0.93989 | 0.93324 | 0.64623 |
| Bi-LSTM | RMSE | 0.01824 | 0.02131 | 0.02287 | 0.02372 | 0.03276 | 0.03586 |
| | MAE | 0.01213 | 0.01669 | 0.01690 | 0.01844 | 0.02561 | 0.03080 |
| | MAPE (%) | 2.97 | 4.07 | 4.48 | 5.48 | 8.22 | 10.93 |
| | R-squared | 0.98791 | 0.97781 | 0.97388 | 0.97042 | 0.93274 | 0.61686 |
| GRU | RMSE | 0.02131 | 0.02437 | 0.02707 | 0.02974 | 0.03616 | 0.04014 |
| | MAE | 0.01648 | 0.01898 | 0.02194 | 0.02612 | 0.03285 | 0.03613 |
| | MAPE (%) | 4.61 | 5.57 | 6.35 | 8.36 | 10.52 | 12.84 |
| | R-squared | 0.97803 | 0.97176 | 0.9634 | 0.95351 | 0.91804 | 0.52475 |
| iTransformer | RMSE | 0.02391 | 0.02657 | 0.03627 | 0.03388 | 0.0453 | 0.08681 |
| | MAE | 0.01913 | 0.02312 | 0.03517 | 0.02930 | 0.0397 | 0.08456 |
| | MAPE (%) | 5.23 | 7.40 | 9.54 | 9.01 | 14.33 | 27.49 |
| | R-squared | 0.97024 | 0.94663 | 0.89173 | 0.91354 | 0.38875 | −1.24507 |
| TSMixer | RMSE | 0.0263 | 0.02877 | 0.03712 | 0.05158 | 0.05621 | 0.09795 |
| | MAE | 0.02175 | 0.02703 | 0.0328 | 0.04392 | 0.05035 | 0.09175 |
| | MAPE (%) | 6.04 | 6.49 | 10.32 | 14.91 | 19.16 | 30.13 |
| | R-squared | 0.96655 | 0.92959 | 0.93121 | 0.86016 | 0.25874 | −1.85807 |

## Verification of feature importance

Feature importance is a metric that indicates the influence of each input feature on the model's predictions. When the model receives input data to make predictions, it is crucial to assess how important each feature is in making those predictions. This information helps enhance the interpretability of the model in deep learning and machine learning.

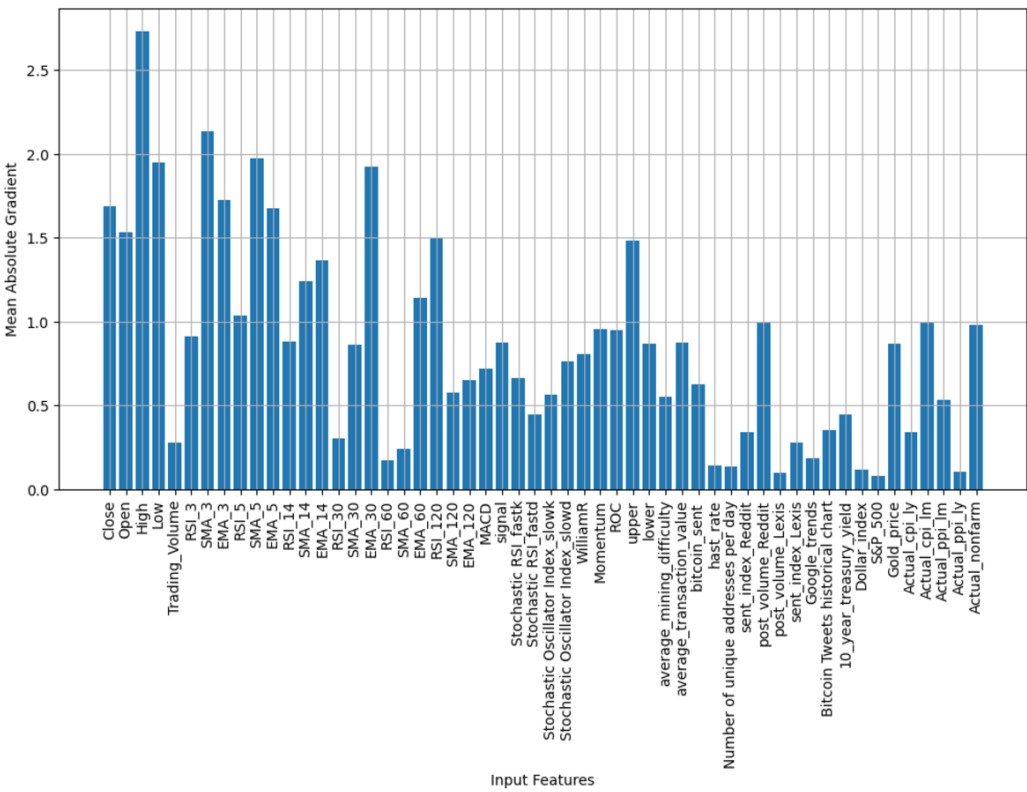

**Figure 16** Feature importance employing mean absolute gradient values.

There are several methods to evaluate the importance of input features, and one such method utilizes gradients. By computing gradients for a given model and input data, the authors obtained the gradients of the loss function with respect to each input feature. These gradients indicate how significantly each feature contributes to the loss function, and a higher gradient value suggests that the feature has a more significant impact on the model's predictions. The authors examined the gradient importance to assess the importance of the input features employed (Fig. 16).

Based on the results, the authors observed the following influences on the prediction of Bitcoin prices across different categories.

Initially, historical Bitcoin price data proved to be the most influential factor in predicting its future values. In terms of technical indicators, short-term technical indicators had a greater influence on predictions compared to long-term indicators. Specifically, the Open price and the upper Bollinger Bands values significantly affected the predictions. Regarding on-chain data, the actual amount of Bitcoin transferred on the blockchain and the average transaction value had a considerable influence. For sentiment-based data, the volume of posts on both Reddit and Twitter had the highest impact on Bitcoin prices. This suggests that the number of posts reflects public interest, which in turn affects price fluctuations.

When examining traditional assets, the price of gold had a substantial influence on Bitcoin prices. Furthermore, macroeconomic indicators generally had a stronger influence

**Table 8  Ablation test based on input features (MAPE, %).**

| Input features | Bi-LSTM |
|---|---|
| Only price data | 5.21 |
| Price data with technical indicators | 3.26 |
| Price data with sentiment index | 3.24 |
| Price data with on-chain data | 3.53 |
| Price data with traditional assets | 5.63 |
| Price data with macroeconomic factors | 6.56 |
| All inputs | **2.97** |

compared to traditional assets, with monthly CPI and PPI demonstrating more influence relative to their annual counterparts. Lastly, non-farm employment indicators also played a pivotal role in shaping Bitcoin price trends.

These findings could contribute to understanding the multifaceted factors influencing Bitcoin price predictions across various domains.

### Feature ablation test

Subsequently, an ablation test was performed to confirm the efficacy of the input features. Utilizing sentiment, technical, and on-chain data together yielded better performance than using only price data. However, the inclusion of traditional assets or macroeconomic data did not enhance the predictive accuracy. It was established that Bi-LSTM achieved its highest performance when all three inputs were incorporated. The experimental results for the MAPE values are listed in Table 8, and a graphical representation of these results is shown in Fig. 17.

### DISCUSSION

This study examined different input window sizes and frameworks, incorporating a wide array of factors grouped into five categories, all presumed to impact Bitcoin prices. The findings revealed that Bi-LSTM produced superior outcomes with a window size of three, with RMSE of 0.01824, MAE of 0.01213, MAPE of 2.97%, and an R-squared value of 0.9879,1 suggesting that short-term price prediction aligns better with Bitcoin's volatile nature than long-term prediction. Interestingly, the performance of the Bi-LSTM was better than the iTransformer or TSMixer, which are commonly referred to as SOTA.

Speculating on the reasons why these SOTA models showed inferior performance compared to Bi-LSTM on Bitcoin price might lead to the following conclusions. The TSMixer involves stacking time-linear models and utilizing cross-variable feed-forwards, encompassing continuous time patterns, cross-variable information, and auxiliary features. It is characterized by its effectiveness in learning time patterns under the assumption of temporal dependencies commonly present. And the iTransformer, as a transformer-based model, may be susceptible to overfitting or poor generalization, particularly when dealing with high uncertainty, leading to a rapid decrease in prediction reliability. These models demonstrate significant strength in capturing recurring shapes such as periodicity.

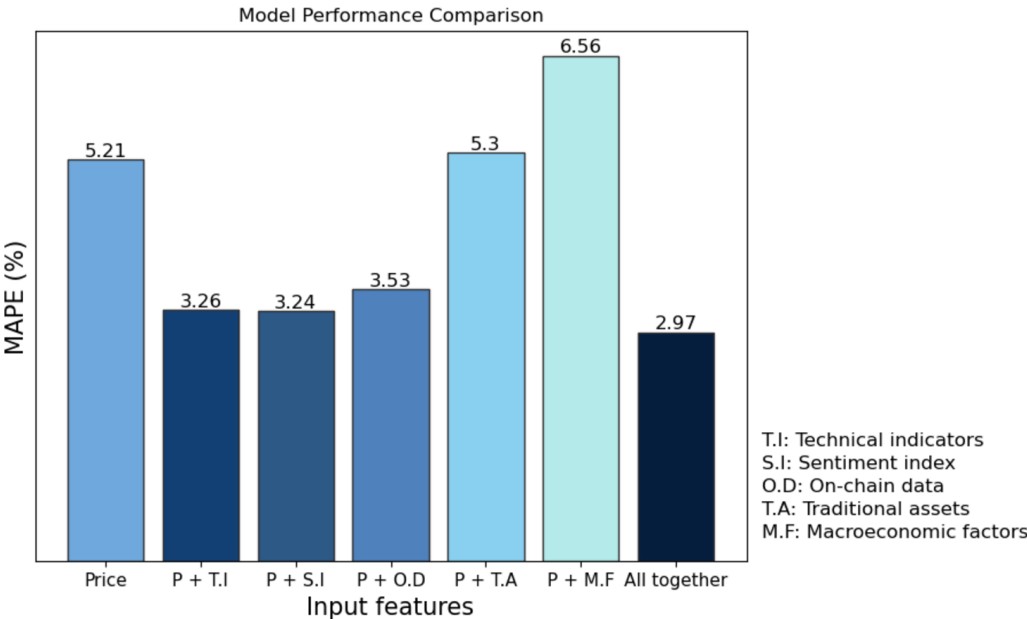

**Figure 17  Visualization of ablation test results.**

However, they may not be suitable for time-series data that lack consistent patterns and exhibit high volatility. In such cases, appropriate denoising techniques may be necessary.

Furthermore, the authors aimed to assess the feature importance for the Bi-LSTM model with a window size of 3, evaluating how each feature contributes to predictions. The authors computed gradients for the given model and input data and visualized these results. Given the nature of price prediction, historical Bitcoin price data had the most significant impact on predictions, and analysis for the remaining four categories was intuitively analyzed through visualization. This experimental process can contribute to understanding the complex factors influencing Bitcoin price predictions.

Moreover, consolidating all inputs for the model input, as observed in the ablation testing, outperformed individual element integration. This result indicated that the input features selected for the experiment were efficient in predicting Bitcoin prices. In conclusion, the significance lies in achieving a respectable performance without separately removing the 2020 COVID-19 outbreak black swan as an outlier by employing diverse input features. In other words, the authors demonstrated efficiency in detecting and predicting assets with volatility characteristics.

Consequently, the findings of this study could ultimately assist in managing extreme volatility and unexpected market changes. Furthermore, by providing a varied examination of the factors influencing price formation, it enables investors to make informed decisions regarding Bitcoin-related investments and helps policymakers legislate considering these factors.

## CONCLUSION

This study demonstrates that Bi-LSTM outperforms SOTA models like iTransformer and TSMixer in predicting Bitcoin prices, particularly with a short-term window size of three, aligning with Bitcoin's volatile nature. The findings suggest that while SOTA models excel at identifying recurring patterns, they struggle with the high volatility and lack of consistent patterns in Bitcoin data. The research underscores the efficiency of using diverse input features collectively, achieving strong predictive performance without excluding outliers like the 2020 COVID-19 outbreak.

However, some limitations must be addressed in future studies. First, comparative analysis with previous studies was limited by a lack of confidence in the reproducible models and data. Future research could consider a trading strategy that uses the framework proposed in this study for validation. Therefore, future research could benefit from developing and validating a trading strategy for profitability assessment using the framework proposed in this study. In other words, incorporating a trading strategy into the proposed framework for real-world validation remains a critical next step. Furthermore, there is a limitation in that there is currently no prominent methodology that considers state-of-the-art unsupervised text sentiment analysis. Future studies need to explore integrating advanced unsupervised sentiment analysis techniques to better capture the sentiment impact on Bitcoin prices. Furthermore, methodologies for analyzing macroeconomic-related text documents, such as speeches or writings by individuals involved in monetary policy, including Federal Reserve officials, would provide deeper insights into market movements. In addition, future research can analyze complexity by applying variational mode decomposition (VMD). For instance, identifying periodic patterns that occur over a specific time range can improve the accuracy of a price prediction model. Finally, adopting a method that applies a variable window size by understanding the relative amplitude of volatility could potentially improve prediction performance. Future research could explore adaptive window sizes that respond to the dynamic nature of Bitcoin's market volatility. In summary, while this study highlights the strengths of Bi-LSTM in predicting Bitcoin prices among high volatility, addressing these limitations and exploring the suggested future research directions will further enhance the robustness and applicability of predictive models in the financial domain.

## ACKNOWLEDGEMENTS

We would like to thank Editage for English language editing.

### Funding

This study was supported by a National Research Foundation of Korea (NRF) grant funded by the Korean government (RS-2023-00208278). The funders had no role in study design, data collection and analysis, decision to publish, or preparation of the manuscript.

## Grant Disclosures

The following grant information was disclosed by the authors:
National Research Foundation of Korea: RS-2023-00208278.

## Competing Interests

The authors declare there are no competing interests.

## Author Contributions

- Hae Sun Jung conceived and designed the experiments, performed the experiments, analyzed the data, performed the computation work, prepared figures and/or tables, authored or reviewed drafts of the article, and approved the final draft.
- Jang Hyun Kim conceived and designed the experiments, authored or reviewed drafts of the article, and approved the final draft.
- Haein Lee conceived and designed the experiments, performed the experiments, analyzed the data, performed the computation work, prepared figures and/or tables, authored or reviewed drafts of the article, and approved the final draft.

## Data Availability

The data and code are available at GitHub and Zenodo:

– https://github.com/Haein34/Bitcoin-Price-Prediction/tree/publish.

– Haein Lee. (2024). Haein34/Bitcoin-Price-Prediction: Bitcoin-Price-Prediction (publish). Zenodo. https://doi.org/10.5281/zenodo.12776628.

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
