# Peer review of "Decoding Bitcoin: leveraging macro- and micro-factors in time series analysis for price prediction"

_PeerJ Computer Science, doi:10.7717/peerj-cs.2314_

## Round 0.1 · original submission · Major Revisions

All three reviewers have taken a positive view of the manuscript, which suggests this work might be publishable in PeerJ Computer Science. Reviewer 3, in particular, asks to expand some sections and provide some more background and details throughout the paper.

·

Basic reporting

Structure conforms to PeerJ standards, discipline norms, or any deviations are to improve clarity.

The paper shows promise, but it requires revisions to meet the journal's standards.

Experimental design

Provide a thorough description of data collection methods.

Modify Figure 1, Creating an effective experimental flowchart for predicting Bitcoin prices involves outlining the entire process clearly and logically.

The experimental design is given in Figure 1, but the flow of work is not clear for that reason I
also raise comment to modify Figure 1 in the Experimental design section.

Validity of the findings

The preprocessing steps are vague or incomplete.

Elaborate on each preprocessing step, including specific methods for handling missing values (e.g., imputation techniques), outlier detection methods, and normalization techniques. Provide examples of feature engineering.

The authors of the manuscript incorporated a comprehensive set of macroeconomic indicators, on-
chain data revealing transactional details.

In the manuscript interestingly the Bi-LSTM outperformed the SOTA models; and in terms of
performance.

Bi-LSTM processes the input sequence in both forward and backward directions, allowing the
model to understand the full context of the data. This is particularly useful for time series data
where future events can be influenced by past trends and vice versa.

This methodology provides a varied examination of the factors influencing price statistics, helping
investors make informed decisions regarding Bitcoin-related investments.

Additional comments

The research work reported is interesting. There, however, are some concerns that the authors need to address.

Reviewer 2 ·

Basic reporting

Please highlight them in your introduction in the specific paragraph.

For ORIGINAL RESEARCH PAPER
* * *
There are four (4) types of novel technical results:
1) An algorithm;
2) A system construct: such as hardware design, software system, protocol,
etc.; The main goal of your revised paper is to ensure that the next person
who designs a system like yours doesn't make the same mistakes and takes
advantage of some of your best solutions. So make sure that the hard
problems (and their solutions) are discussed and the non-obvious mistakes
(and how to avoid them) are discussed;
3) A performance evaluation: obtained through analyses, simulation or
measurements; or
4) A theory: consisting of a collection of theorems.
Your final camera ready paper should focus on:
1) Describing the results in sufficient details to establish their validity;
2) Identifying the novel aspects of the results, i.e., what new knowledge is
reported and what makes it non-obvious; and
3) Identifying the significance of the results: what improvements and impact
do they suggest.

Experimental design

1. Methods and Replication:
Accepted

2. Data Preprocessing: Accepted
The paper includes a section on feature engineering, which discusses the preprocessing of data into features for input into the framework.

3. Experiment Results :
To enhance the clarity and comprehensiveness of the results presentation, the following improvements could be considered:

a. Detailed Descriptions: Provide more detailed descriptions of the model training processes, including hyperparameter tuning, data preprocessing steps, and any challenges encountered during the modeling process. This additional context can help readers understand the nuances of the results.

b. Expanded Comparisons: While the comparison between models is clear, expanding this section to include a more detailed discussion on why certain models performed better or worse could provide deeper insights. Discussing the strengths and weaknesses of each model in the context of the specific dataset used would add value, and add a comparison of the data splitting process 80:20, 70:30 , 90:10 to justify which experiment is better , and for the last and why choose Experimental Results Based on the Difference in Window Size rather than testing using many epochs ?

c. Integration of Factors:

The methodology for integrating macroeconomic indicators, on-chain data, and sentiment analysis may lack clarity. A detailed explanation of how these diverse data sources are combined in the model is necessary to understand the interactions and contributions of each factor.

d, Volatility Handling:
While the paper successfully demonstrates the effectiveness of short-term windows, it does not extensively address how to manage extreme volatility and sudden market shifts, especially considering the impact of unforeseen events like the COVID-19 pandemic.

e. Expand Assessment Metrics:

Assessment Metrics:

Using only MAPE does not provide a comprehensive view of the model's accuracy, as it might not handle zero or near-zero actual values well and can be skewed by outliers.
The paper does not mention the use of other relevant metrics like Root Mean Squared Error (RMSE) or Mean Absolute Error (MAE), which could provide additional insights into the model performance.

Use additional metrics such as RMSE, MAE, and R-squared to provide a more comprehensive evaluation of the model's predictive accuracy and robustness.
Consider evaluating the model's performance on different types of errors (e.g., systematic vs. random errors) to understand its strengths and weaknesses better.

Validity of the findings

1. Window Size Analysis:

Although the paper investigates various window sizes (short-term, mid-term, long-term), the explanation of how these window sizes impact prediction performance is somewhat limited. More detailed analysis and discussion on why certain window sizes perform better than others could enhance understanding

2.Model Comparison:

While the paper compares several models (CNN, RNN, LSTM, etc.), it does not provide a comprehensive comparison against baseline models or simpler statistical methods. Including such comparisons would highlight the advantages and limitations of advanced models more clearly​​.

3. Impact of External Factors:

The incorporation of sentiment analysis, news sources, and Google Trends is a strong point, but the impact of these external factors is not deeply explored. A more in-depth analysis of how these factors influence Bitcoin price predictions could provide more actionable insights

4. Conlusion : The conclusion of the paper does identify unresolved questions, limitations, and future directions.

Reviewer 3 ·

Basic reporting

The paper is very interesting, and the topic is relevant. It certainly has the potential to contribute to the field, however, some issues must be addressed.
1. Please provide the most important numerical results briefly at the end of abstract.
2. Introduction should be restructured. It is too short, and you should consider to provide the following information:
State the research gap clearly in the Introduction, to justify the need for your approach.
Moreover, you should highlight the main contributions clearly as a bulleted list in the Introduction.
Lastly, provide paper structure at the end of the Introduction.
3. Related section should be improved, as numerous recent important papers that deal with cryptocurrencies forecasting are missing, like:
Mizdrakovic, Vule, et al. "Forecasting bitcoin: Decomposition aided long short-term memory based time series modelling and its explanation with shapley values." Knowledge-Based Systems (2024): 112026.
Todorovic, Mihailo, et al. "Multivariate Bitcoin price prediction based on LSTM tuned by hybrid reptile search algorithm." 2023 16th International Conference on Advanced Technologies, Systems and Services in Telecommunications (TELSIKS). IEEE, 2023.
Gupta, Ruchi, and Jagannath E. Nalavade. "Metaheuristic assisted hybrid classifier for bitcoin price prediction." Cybernetics and Systems 54.7 (2023): 1037-1061.
Strumberger, Ivana, et al. "Multivariate Bitcoin Price Prediction Based on Tuned Bidirectional Long Short-Term Memory Network and Enhanced Reptile Search Algorithm." International Conference on Information and Software Technologies. Cham: Springer Nature Switzerland, 2023.
Behera, Sudersan, Sarat Chandra Nayak, and AVS Pavan Kumar. "Evaluating the performance of metaheuristic based artificial neural networks for cryptocurrency forecasting." Computational Economics (2023): 1-40.
Salb, Mohamed, et al. "Support vector machine performance improvements for cryptocurrency value forecasting by enhanced sine cosine algorithm." Computer Vision and Robotics: Proceedings of CVR 2021. Singapore: Springer Singapore, 2022. 527-536.

4. Avoid using personal nouns (like we/our) throughout the paper.
5. Thorough proofreading is recommended.

Experimental design

1. Regarding metrics, discuss in more details why you use MAPE, and not MSE to evaluate the models.
2. Make sure that each parameter in every equation has been explained in the text. For example, MAPE equation parameters were not discussed.
3. Also, each equation should be numerated, and you should refer to it from the text (for example, MAPE calculation is provided in Eq. (2)).
4. Did you consider using VMD (variational mode decomposition)?
5. Make sure to provide enough details about the simulation setup to support replication studies.
6. Discussion is very brief, it should be more elaborate.

Validity of the findings

1. Did you consider utilizing explainable AI (like SHAP)? It could be an excellent addition to the paper, and help in understanding feature importance and influence on predictions.
2. Discuss the limitations of the proposed approach in more details.

---

## Round 0.2 · Minor Revisions

There is only one outstanding comment, which seems rather minor.

·

Basic reporting

#no comment
The authors have improved the work by providing more details.

Experimental design

#no comment
The authors have specifying the overall experimental design.

Validity of the findings

#no comment

Additional comments

The authors have added more detailed analysis, including tables and figures that effectively illustrate their findings.

Reviewer 2 ·

Basic reporting

Intro & background to show context. Literature well referenced & relevant.

Experimental design

1. I couldn't find an exact mention of the impact of window size on book Box, G. E. P., Jenkins, G. M., Reinsel, G. C., & Ljung, G. M. (2015)
based on
"Window size indicates the extent of historical data considered when making predictions for a given period (Box et al., 2015). Experiments exploring the effectiveness of various input window sizes for the short-term (3 and 5 days), mid-term (14 and 30 days), and long-term (60 and 120 days) were executed to identify the most effective window size for predicting Bitcoin prices. Regardless of the input window size, the output window size was fixed for a single day (Figure 7)."

Please explain more carefully and in more detail if you really took it from the book.
for example from where this equation window sizes = 3 equals 3 days ?

Validity of the findings

Conclusions are well stated & limited to supporting results.

Reviewer 3 ·

Basic reporting

The authors have revised the paper according to the comments from the previous round. The paper is clear and has a good structure.

Experimental design

Methods are properly described. Evaluation methods are appropriate.

Validity of the findings

The experiments and evaluations are performed properly. The conclusions are well stated. Limitations of the study are discussed.

---

## Round 0.3 · accepted · Accept

The reviewers are happy with the current version of the manuscript, which can therefore be accepted for publication in PeerJ Computer Science.

Reviewer 2 ·

Basic reporting

Intro & background to show context. Literature well referenced & relevant.

Experimental design

Rigorous investigation performed to a high technical & ethical standard.

Validity of the findings

Conclusions are well stated & limited to supporting results.